# Clinical Perspectives of ERCC1 in Bladder Cancer

**DOI:** 10.3390/ijms21228829

**Published:** 2020-11-22

**Authors:** Konstantinos Koutsoukos, Angeliki Andrikopoulou, Nikos Dedes, Flora Zagouri, Aristotelis Bamias, Meletios-Athanasios Dimopoulos

**Affiliations:** 1Department of Clinical Therapeutics, Alexandra Hospital, Medical School, 11528 Athens, Greece; koutsoukos.k@gmail.com (K.K.); aggelikiandrikop@gmail.com (A.A.); dedes.nik.95@gmail.com (N.D.); florazagouri@yahoo.co.uk (F.Z.); 22nd Propaedeutic Department of Internal Medicine, Medical School, National and Kapodistrian University of Athens, “ATTIKON” University Hospital, Rimini 1, 12462 Chaidari, Greece; abamias@med.uoa.gr

**Keywords:** ERCC1, excision repair cross-complementation group 1, bladder cancer, urothelial, biomarker

## Abstract

ERCC1 is a key regulator of nucleotide excision repair (NER) pathway that repairs bulky DNA adducts, including intrastrand DNA adducts and interstrand crosslinks (ICLs). Overexpression of ERCC1 has been linked to increased DNA repair capacity and platinum resistance in solid tumors. Multiple single nucleotide polymorphisms (SNPs) have been detected in ERCC1 gene that may affect ERCC1 protein expression. Platinum-based treatment remains the cornerstone of urothelial cancer treatment. Given the expanding application of neoadjuvant and adjuvant chemotherapy in locally advanced bladder cancer, there is an emerging need for biomarkers that could distinguish potential responders to cisplatin treatment. Extensive research has been done regarding the prognostic and predictive role of ERCC1 gene expression and polymorphisms in bladder cancer. Moreover, novel compounds have been recently developed to target ERCC1 protein function in order to maximize sensitivity to cisplatin. We aim to review all the existing literature regarding the role of the ERCC1 gene in bladder cancer and address future perspectives for its clinical application.

## 1. Introduction

Worldwide, urothelial cancer is the 10th most common malignancy in both sexes accounting for 3% of all new cancer cases [1]. Based on GLOBOCAN data, urothelial cancer affected 549,393 individuals worldwide in 2018, while 199,922 total deaths were reported. Despite novel treatment strategies, 5-year survival rate does not exceed 70% for muscle-invasive bladder cancer (MIBC) and 5% for metastatic disease in the US [2]. Invasion of the detrusor muscle of the bladder constitutes a major contributor to disease prognosis, as 5-year survival rates reach 96% in the non-invasive population [2]. Initially 20% of patients will present with MIBC, while 15–20% of non-muscle-invasive bladder cancer (NMIBC) will eventually expand to the muscle layer [3]. Management of NMIBC is based on transurethral resection of the bladder tumor (TURBT) and intravesical therapy, like bacillus Calmette–Guérin (BCG) which leads to a 37% further reduction in recurrence rate [4]. In case of muscle invasion, however, radical cystectomy (RC) and pelvic lymph node dissection remain the cornerstone of treatment. For MIBC patients unfit for cystectomy, a bladder-preserving approach based on concurrent chemoradiotherapy can be adopted. Neoadjuvant chemotherapy (NAC) has been incorporated in current clinical practice based on Phase III studies and a meta-analysis of 11 clinical trials showing a 5-year OS benefit of 5% [3,4,5]. Cisplatin-based adjuvant chemotherapy remains a controversial issue that should be personalised in high-risk MIBC patients with extravesical or node-positive disease or positive margins [3,4,5]. Current data reveal a significant prolongation in progression-free survival (PFS), however the contribution to overall survival (OS) is unclear [3,6]. As for locally advanced or metastatic disease, platinum-based treatment regimens remain the gold-standard of treatment.

Given the limited benefit of neoadjuvant and adjuvant cisplatin-based chemotherapy, there is a rising need for biomarkers that could predict response to cisplatin-based chemotherapy. The development of biomarkers like genetic variants and gene products could help to distinguish the subgroup of patients with bladder cancer that will benefit more from platinum chemotherapy and optimize clinical practice. ERCC1, ERCC2, ATM, FANCC and RB1 are some of the genes associated with platinum sensitivity and clinical outcome. Excision repair cross-complementing 1 (ERCC1) is a protein that has been related to response to platinum-based chemotherapy in multiple cancer types [7,8,9]. Testicular germ cell tumors (TGCT) are characterized by increased sensitivity to cisplatin compared to the other cancer subtypes, demonstrating an 80% response rate [10]. It was recently shown that this hypersensitivity of testicular cells derives from reduced ERCC1–XPF expression and thus limited interstrand crosslink (ICL) repair ability [10]. ERCC1 has emerged as a novel biomarker of bladder cancer over the years and could be potentially targeted to maximize platinum sensitivity.

ERCC1 is a 15.3-kb gene located on 19q13.2-q13.3 that encodes a 33-kDa protein composed of 297 amino acids. ERCC1 protein interacts with xeroderma pigmentosum group F (XPF) through their C-terminal domains and form a heterodimeric protein complex with an endonuclease activity. ERCC1–XPF complex is one of the key components of the nucleotide excision repair (NER) pathway by catalyzing incision on the 5′-side of the damaged DNA strand [11]. There are six major DNA repair pathways in human cells: base excision repair (BER), nucleotide excision repair (NER), mismatch repair (MMR), homologous recombination (HR), nonhomologous endjoining (NHEJ), and translesion DNA synthesis (TLS) [12]. The NER pathway acts on a variety of helix-distorting DNA lesions to remove UV-induced pyrimidine dimers, photo-products, chemical adducts and cross-links [12]. Environmental and occupational chemical carcinogens, such as hydrocarbons, arylamines, nitrosamines lead to DNA adduct formation mostly repaired by the NER pathway. During NER, xeroderma pigmentosum group A (XPA) and replication protein A (RPA) are recruited to the damaged site to enable the stabilization of the NER complex to the unwound DNA intermediate [13]. The ERCC1–XPF protein complex interacts with both RPA and XPA to make the 5’ incision. Apart from endonuclease activity in the NER pathway, the ERCC1–XPF complex implicates other repair pathways as well, including interstrand crosslink repair (ICL) and both homologous recombination repair and non-homologous end-joining of DNA double-strand breaks (DSBs) [13]. The ERCC1–XPF complex catalyzes the incision on either side of the interstrand crosslink to release one arm of the crosslink and initiate repair [14]. Cisplatin is an alkylating agent that interferes with DNA replication via forming DNA adducts, which include intrastrand DNA adducts and DNA interstrand crosslinks. Unrepaired lesions result in the inhibition of DNA replication and transcription, activating cell apoptosis. Therefore, increased repair of intra- and interstrand DNA adducts due to ERCC1–XPF overexpression could prove detrimental for cisplatin-induced cell death. Inversely, the inhibition of ERCC1–XPF complex would sensitize cancer cells to cisplatin and other regiemens whose effect requires ERCC1–XPF repair activity. 

Urothelial cancer fate is particularly affected by the development of cisplatin resistance. ERCC1 and other DNA repair gene alterations could attain a dual role in urothelial cancer management: as a biomarker defining urothelial cancer prognosis and as a selection tool to identify patients that will benefit more from cisplatin treatment. In this article, we review all existing data regarding the clinical application of ERCC1 and other repair genes in urothelial cancer. 

## 2. ERCC1 in Bladder Cancer

Several studies demonstrated no association between ERCC1 expression and sex, age and ECOG performance status clinical and pathological grading [15,16,17,18,19,20]. One study reported a higher ERCC1 expression in metastatic tumors compared with primary (*p* = 0.066) [19]. The percentage of ERCC1 high expressing tumors is controversial, since there is no prespecified positivity threshold. The existing literature supports that high ERCC1 expression is related to a more favorable outcome in MIBC treated with surgery alone. This finding could be attributed to reduced DNA repair capacity and greater tumor mutational burden in ERCC1-low expressing tumor cells. Klatte et al. investigated the prognostic and predictive role of ERCC1 in 432 patients treated with radical cystectomy [17]. In the absence of adjuvant chemotherapy, ERCC1-positive MIBC patients had a significantly reduced risk of recurrence on univariate and multivariate analysis (*p* = 0.021 and *p* = 0.028) [17]. Indeed, patients with ERCC1-positive tumors had a 5-year disease-free survival (DFS) rate of 62% compared to 49% of patients with ERCC-negative tumors. The same relationship was observed between ERCC1-negative tumors and cancer-specific survival (*p* = 0.032) [17]. Sun et al. conducted a retrospective study of ERCC1 expression in 93 patients treated with radical cystectomy with or without cisplatin-based adjuvant chemotherapy [15]. In the subpopulation not treated with adjuvant chemotherapy ERCC1 positivity was associated with a significantly prolonged survival (*p* = 0.049). ERCC1-positive patients reported an 84% five-year overall survival rate compared to 49.2% of ERCC1-negative patients (*p* = 0.083) [15]. Hemdan et al. also reported a worse prognosis in ERCC1-negative patients treated with cystectomy without neoadjuvant chemotherapy (*p* = 0.005) [21]. In upper tract urothelial carcinoma, however, ERCC1 expression was not predictive of overall survival (*p* = 0.48) or cancer-specific survival (*p* = 0.33) in patients that underwent radical nephroureterectomy [20]. Overall, low ERCC1 expression may lead to reduced DNA repair capacity and increased genomic instability. Inversely, ERCC1-positive tumors are characterized by enhanced DNA repair ability and survival benefit against ongoing mutagenesis. ERCC1 positivity could serve as a favorable prognostic indicator in the absence of platinum-based chemotherapy. 

## 3. ERCC1 Expression and Neoadjuvant Treatment

ERCC1 expression has been extensively studied regarding its role in the identification of responders to neoadjuvant treatment (Table 1). Ozcan et al. demonstrated that high ERCC1 expression is associated with shorter disease-free survival (DFS) (*p* = 0.019) and overall survival (OS) (*p* = 0.002) in MIBC patients receiving platinum-based neoadjuvant treatment [22]. Pathologic complete response rate was similar between ERCC1 low- and high-expressing tumors. Based on this study, high ERCC1 expression could serve as an indicator of inferior outcome in terms of DFS and OS. As was stated previously, Hemdan et al. demonstrated that negative ERCC1 expression is a poor prognostic marker for patients treated only with cystectomy. In neoadjuvant pretreated population, however, the ERCC1-negative population exhibited a prolonged survival (*p* = 0.002) compared with ERCC1-positive one [21]. In other words, ERCC1-negative BC patients would benefit more from the addition of neoadjuvant chemotherapy prior to surgical intervention. Eldehna et al. assessed ERCC1 expression in 80 non-operable, nonmetastatic MIBC patients who received 4–6 cycles of platinum-based chemotherapy [23]. Negative ERCC1 expression was associated with increased treatment response (*p* = 0.013) [23]. However, there was no relationship between ERCC1 expression and mean progression-free survival (PFS) (0.794) or OS (*p* = 0.499). Therefore, Eldehna et al. proposed the use of ERCC1 expression as a predictive rather than a prognostic biomarker for neoadjuvant chemotherapy administration. The co-expression of Snail and ERCC1 but not ERCC1 alone was associated with shorter DFS and OS in patients receiving neoadjuvant chemotherapy (*p* = 0.029 and *p* = 0.040) [24]. Of note, there was a profound relationship between expression of ERCC1 and Snail (*p* = 0.001). The co-expression of Snail and ERCC1 has been previously identified as a poor prognostic factor for cisplatin-treated head and neck carcinomas, further supporting this evidence [25]. Necchi et al. associated ERCC1 expression and response to cisplatin-based neoadjuvant treatment in a Phase II study evaluating addition of sorafenib to the neoadjuvant setting [26]. The translational analysis demonstrated that ERCC1 expression is significantly related to pathologic response (*p* = 0.047). Choueiri et al. reported no statistically significant association between ERCC1 expression and pathologic response or DFS in patients administered with four cycles of neoadjuvant dose-dense methotrexate, vinblastine, doxorubicin and cisplatin (ddMVAC) [27]. However, the sample size was limited. Yang et al. correlated high ERCC1 expression with increased response to neoadjuvant chemotherapy (*p* = 0.011), although this is the only study reporting contradictory results to the aforementioned studies [28]. Overall, there is a pattern of low ERCC1 expression in patients responding to neoadjuvant chemotherapy. 

## 4. ERCC1 Expression and Adjuvant Chemotherapy

As stated earlier, Sun et al. confirmed the prognostic and predictive value of ERCC1 expression in patients undergoing radical cystectomy with adjuvant chemotherapy (Table 2) [15]. In the arm receiving adjuvant chemotherapy after radical cystectomy, ERCC1 positivity was associated with reduced survival (*p* = 0.047), whereas ERCC1 negativity was significantly linked to survival benefit from adjuvant chemotherapy in terms of DFS and overall survival (*p* = 0.20 and *p* = 0.034, respectively). The five-year overall survival was 41.6% for ERCC1-positive MIBC patients and 71.8% for ERCC1-negative patients. Hoffmann et al. investigated multidrug resistance gene 1 (MDR1) and ERCC1 gene expression in 108 patients with locally advanced bladder cancer that received cisplatin-based adjuvant chemotherapy [29]. Low ERCC1 expression was associated with prolonged PFS (*p* = 0.01). Indeed, 45% of patients with ERCC1 low-expressing tumors reported disease progression within five years compared with 70% of patients with ERCC1 high-expressing tumors. Median overall survival was also longer for ERCC1 low-expressing patients (72.4 vs. 33.1 months), although not statistically significant (*p* = 0.19) [29]. The present study further established the role of ERCC1 as a predictive marker of platinum-based adjuvant chemotherapy. Klatte et al. found no association between ERCC1 and response to cisplatin (*p* = 0.88), although they confirmed the prognostic role of ERCC1 expression to disease-free and cancer-specific outcome [17].

## 5. ERCC1 Gene in the Locally Advanced/Metastatic Setting

Most existing evidence regarding ERCC1 role in urothelial cancer emerged from studies assessing metastatic or locally advanced disease (Table 3). Bellmunt et al. associated low ERCC1 expression to prolonged OS in locally advanced or metastatic urothelial cancer patients treated with cisplatin-based chemotherapy [30]. Overall survival was 25.47 and 15.4 months in ERCC1 low-expressing and high-expressing tumors respectively, indicating a survival benefit in favor of low ERCC1 expression [30]. High ERCC1 expression as quantified by RT-PCR was associated with a worse outcome (*p* = 0.012), while median time to progression was prolonged in ERCC1 low-expressing tumors (*p* = 0.087). The same results were confirmed by immunohistochemistry assay (IHC) [31]. Median disease-specific survival was 12.6 months in ERCC1-negative patients compared with 8.6 months in ERCC1-positive ones (*p* = 0.032) [31]. These two studies confirmed the prognostic role of ERCC1 expression in cisplatin-treated advanced or metastatic urothelial cancer. Matsumara et al. evaluated human equilibrative nucleoside transporter 1 (hENT1) and ERCC1 expression in patients with metastatic urothelial cancer treated with first line cisplatin–gemcitabine chemotherapy [32]. No statistically significant difference was observed in PFS (10.7 and 13.0; *p* = 0.33) and median survival (13.6 vs. 17.1; *p* = 0.178) between the ERCC1 high- and ERCC1 low-expressing groups. However, when jointly low hENT1 and high ERCC1 tumors were compared to high-hENT1- and low-ERCC1-expressing ones a shorter median survival time was observed (11.6 vs. 19.9 months) [32]. Kim KH et al. demonstrated a statistically significant prolongation of PFS in ERCC1-negative metastatic urothelial cancer patients that received cisplatin-based chemotherapy (10.6 vs. 8.4 months, *p* = 0.03) [33]. However, the difference in median OS between the two populations was not statistically significant (*p* = 0.73). Low ERCC1 expression was identified as a prognostic biomarker of longer OS in cisplatin-treated metastatic urothelial carcinoma in another study, especially when combined with low RAD51 expression (*p* = 0.0007) [34]. These results enhanced the prognostic role of ERCC1-negative/low expression in metastatic urothelial cancer. Song et al. investigated the role of ERCC1 and BRCA1 genes in forty-two patients with locally advanced or metastatic MIBC that received oxaliplatin plus gemcitabine [16]. The dual-negative BRCA1 and ERCC1 population demonstrated a significantly longer median survival than dual-positive subpopulation (*p =* 0.018). Importantly, the expression of ERCC1 exhibited a significant correlation with positive BRCA1 expression (*p =* 0.044). This correlation was also reported in Bellmunt et al. study (*p* = 0.025) demonstrating an interdependent relationship between the two DNA repair genes [30]. Indeed, ERCC1 is implicated in DNA double-strand break repair apart from the NER pathway, whereas BRCA1 is crucial for NER repair as well. Necchi et al. reported no association between ERCC1 expression and PFS or OS (*p* = 0.750 and *p* = 0.893 respectively) in patients receiving first-line, platinum-based chemotherapy [35], while Kim et al. found no association between PFS or OS and ERCC1 expression in patients with unresectable/metastatic urothelial cancer (PFS: *p* = 0.096, OS: *p* = 0.444) [19]. Urun et al. conducted a meta-analysis including 1475 patients from 13 studies [36]. ERCC1 positivity was significantly associated with worse PFS (*p* = 0.006), however, the association with poor DFS and OS did not reach statistical significance (*p* = 0.09 and *p* = 0.754) [36]. Overall, ERCC1-negative urothelial tumors demonstrate a greater chemosensitivity in the neoadjuvant, adjuvant or metastatic setting.

## 6. ERCC1 and Bladder-Preserving Approach

In vivo studies showed that ERCC1–XPF mutant mice are hypersensitive in gamma irradiation, confirming the role of ERCC1 in DSBs repair [37]. Given the relationship between ERCC1 and sensitivity to radiation, it may be postulated that ERCC1-negative/low-expressing urothelial tumors would benefit more from chemoradiation therapy (CRT) [37]. Moreover, the decision for bladder-preserving treatment based on chemoradiation might exclude the possibility of surgical treatment, so a careful patient selection should be made. Negative ERCC1 expression showed a sensitivity and specificity of 75% and 85.7%, respectively, in predicting response to chemoradiation treatment (*p* = 0.008) [38]. Furthermore, five-year survival was 31.2% and 69.2% in ERCC1-positive and -negative tumors respectively, indicating a favorable outcome in ERCC1-negative cases (*p* = 0.088). Shilkrut et al. explored the clinical significance of ribonucleoside reductase subunit M1 (RRM1) and ERCC1 in thirty-nine MIBC patients that underwent TURBT and platinum- or gemcitabine-based chemoradiation [39]. Complete response rate was lower in ERCC1-positive tumors (67% vs. 84%), although it was not statistically significant (*p* = 0.39). In contrast to these results, Sakano et al. provided evidence that ERCC1 positive expression is related to a better outcome in patients treated with TURBT and platinum-based CRT [18]. Current evidence regarding the role of ERCC1 in urothelial cancer patients undergoing a bladder-preserving approach with combined chemoradiation remains contradicting. Prospective studies should be conducted to address this issue.

## 7. ERCC1 Polymorphisms in Bladder Cancer

ERCC1 gene is composed of 10 exons and multiple gene polymorphisms have been detected that may implicate in carcinogenesis. Multiple ERCC1 variants have been identified and are listed in Table 4. The two most common ERCC1 single-nucleotide polymorphisms (SNPs) are C118T (rs11615) with a T to C substitution (TT to CT/CC) at Exon 4, C8092A (rs321986) with a C to A substitution (CC to CA/AA) in the 3’-untranslated region and 17677A > C (rs3212961) as they may contribute to cancer susceptibility. C118T (rs11615) and C8092A (rs3212986) SNPs have been investigated thoroughly in correlation with the altered risk of several cancer types including lung, ovarian, colorectal and bladder cancer [8,40,41,42]. The rs11615 polymorphism (AAC, AAT) has been linked to a 50% reduction in the transcription level, and thus lower basic levels of ERCC1 and NER complex activity [43]. On the other hand, C8092A (rs3212986) polymorphism is thought to increase the stability of ERCC1 mRNA or be involved in translational repression of ERCC1 mRNA [44]. In contrast to C118T (rs11615) SNP, the presence of the C8092A allele might lead to increased ERCC1 expression and NER activity, and thus greater resistance to platinum treatment. Extensive research has been made on the value of ERCC1 polymorphisms as predictive biomarkers for response to chemotherapy. The CT and TT genotypes in codon 118 (rs11615) were related to reduced risk of disease progression and death compared with CC genotype in ovarian cancer patients [42,45]. C118T (rs11615) polymorphism but not C8092A influences the cancer risk and overall survival of cisplatin-treated NSCLC patients (rs3212986) [40,46]. In colorectal cancer, C118T (rs11615) SNP was associated with worse prognosis (HR, 1.51; 95% CI, 1.01–2.27) [8]. Several studies were conducted to assess ERCC1 polymorphisms as prognostic indicators in bladder cancer.

Nikitas et al. investigated the prognostic role of C118T (rs11615) and C8092A (rs3212986) SNPs in 113 platinum-treated advanced urothelial cancer patients [47]. The homozygous C118T T/T (rs11615) genotype was associated with prolonged cancer-specific survival (*p* = 0.026). Indeed, patients with C118T T/T homozygous genotype had more than two-fold reduction in the risk of death [47]. Matullo et al. conducted a DNA repair gene polymorphism analysis in seven repair genes implicated in different repair pathways in association with bladder cancer risk and smoking [48]. A decreased risk for bladder cancer was observed in C118T variant (CC and CT vs. TT) among smokers (OR: 0.62; 95% CI, 0.41–0.95). In addition, a significant protective effect of XPD/ERCC1-ACC haplotype was found across never smokers (OR = 0.16; 95% CI, 0.03–0.81). Indeed, Ricceri et al. demonstrated that ERCC1-ACC haplotype was significantly decreased in 456 patients with bladder cancer in a case-control study, whereas ERCC1-GAT haplotype (T allele in rs11615) was significantly more present [49]. Ricceri et al. also demonstrated the protective role of three rare SNPs rs967591 (OR = 0.66, CI 95% 0.46–0.95), rs735482 (OR = 0.62, CI 95% 0.42–0.90) and rs2336219 (OR = 0.63, CI 95% 0.43–0.93) in bladder cancer [49]. In another study, Matullo et al. confirmed the driving role of XPD/ERCC1-GAT haplotype in bladder cancer (OR = 1.38; 95% CI = 1.06–1.79) [50]. Closas et al. investigated the role of 22 SNPs in seven NER genes, namely *XPC, RAD23B, ERCC1, ERCC2, ERCC4, ERCC5* and *ERCC6,* on bladder cancer risk. Four SNPs in four of the seven genes were shown to increase susceptibility to bladder cancer compared to wild-type genotypes, including ECCR1 17677A > C (rs3212961) (*p* = 0.06) [51]. Apart from this SNP, a global test for NER pathway polymorphisms proved to significantly predict bladder cancer risk (*p* = 0.04) [51]. The significance of joint associations of different SNPs with bladder cancer prognosis was highlighted by Gu et al. as well [52]. NER pathway polymorphisms, including ERCC1 C8092A (rs3212986) SNP were evaluated in patients with superficial bladder cancer. There was no statistically significant association between NER SNPs and recurrence, apart from an ERCC6 SNP. However, when individual SNPs were combined together there was a trend for increased recurrence risk and shorter recurrence-free interval with increasing number of variants (*p* = 0.0007) [52]. Indeed, patients with six to seven and eight or more high-risk alleles had higher recurrence rates compared with those with five or fewer high-risk alleles (*p* for trend < 0.001). The same pattern was observed in another study, where an increasing number of potential high-risk alleles of NER pathway and other cell-cycle pathways was correlated with increased risk of bladder cancer development [53]. For NER pathway, patients with four, five to six and more than seven high risk alleles had an increased risk of bladder cancer (*p* for trend < 0.001) compared with the reference group of less than four alleles [53]. The same relationship was identified in the combined analysis of DNA-repair and cell-cycle SNPs. Each additional high-risk allele contributes a 1.21-fold increase in risk. The aforementioned studies demonstrated an additive effect of genetic variations in occurrence risk and survival rates in urothelial cancer, rather than a significant impact of an individual SNP.

Addressing the prognostic role of ERCC1 SNPs in response to treatment, Xu et al. demonstrated that patients with an ERCC1 C118T C/C tumor genotype have a significantly improved short-term response to platinum-based chemotherapy (*p* = 0.018) [54]. Consistently, C/C genotype was also associated with improved median PFS (6.3 vs. 4.2 months; *p* = 0.023) and median OS (11.7 vs. 8.5 months; *p* = 0.040) compared with C/T and T/T genotypes. The study indicated ERCC1 C118T (rs11615) SNP as a predictive and prognostic marker in platinum-treated advanced urothelial cancer [54]. A study by Sakano et al. showed that NER pathway SNPs might be prognostic factors for both cancer-specific survival (*p* = 0.04) and treatment-induced toxicity in MIBC treated with chemoradiotherapy [55,56]. However, ERCC1 polymorphisms were not included in these studies. 

## 8. Future Perspectives

Given the essential role of ERCC1–XPF complex in DNA repair and the growing resistance to platinum regimens, ERCC1 consists of an attractive target to increase cisplatin cytotoxicity and efficacy. Indeed, downregulation of XPF-ERCC1 by siRNA transfection impaired repair capacity of cisplatin-induced interstrand and intrastrand crosslinks in NSCLC, ovarian and breast cancer cell lines [57]. XPF-ERCC1 knockdown efficiently sensitized NSCLC to cisplatin [57]. Accordingly, the development of small molecules that target ERCC1, XPF or the related complex could potentially enhance cisplatin toxicity, even in platinum-resistant tumors. A high-throughput screen of chemical compounds revealed two potent ERCC1–XPF complex inhibitors that could be used in conjuction with cisplatin in bladder cancer [58]. 7-hydroxystaurosporine (UCN-01) is a novel G2-checkpoint inhibitor which attenuates the XPA–ERCC1 interaction [59]. UCN-01 suppressed the repair of cisplatin-induced DNA lesions and potentiated cisplatin cytotoxicity, although their clinical application was limited by dose-limiting toxicities in Phase I studies [59,60]. Another compound inhibiting ERCC1–XPF interaction, namely NSC 130813, effectively synergized with cisplatin and mitomycin C in lung cancer and colon cancer cell lines [61]. An alternative way to target the ERCC1–XPF complex is via inhibition of the ERCC1–XPA protein interaction which functions as a scaffold for the localization of ERCC1–XPF to DNA. Research could focus on the identification of small molecule inhibitors that could exploit the ERCC1 contribution to cisplatin resistance. A major future perspective is the development of machine learning (ML)-based gene signatures that could accurately predict response to well-known chemotherapeutic regimens [62]. ERCC1 could be a member of the gene set correlated with the GI50 values of platin drugs, contributing to the overall cross-validation accuracy of a gene signature. Given the multigenetic nature of cisplatin resistance, a combined gene signature would be a more accurate predictor of platinum response than the evaluation of a single gene alone. 

## 9. Other DNA Repair Genes in Bladder Cancer

Alterations in other DNA damage response (DDR) genes implicated in NER and HR repair pathways have been extensively studied apart from ERCC1 gene in urothelial cancer. Mutations in ATM, RB1, and FANCC genes occur in approximately 11%, 14%, and 2% of bladder tumors, respectively [63,64]. Plimack et al. demonstrated that mutations in the DNA repair genes ATM, RB1, and FANCC are associated with increased response to neoadjuvant cisplatin-based chemotherapy in bladder cancer [65]. Mutation in one or more of the three DNA repair genes ATM, RB1, and FANCC predicted pathologic response (87% sensitivity, 100% specificity) and better overall survival (*p* = 0.007). The mutations mostly encountered were deleterious or disrupting protein structures or essential protein interaction sites leading to impaired DNA repair capacity [65]. After a 5-year follow-up period, survival rate was higher in patients with at least one gene mutation (85% vs. 46%) [66]. ERCC2 is a gene involved in NER pathway that is mutated in about 12% of urothelial carcinomas. Data from the same cohort demonstrated that somatic ERCC2 mutations were also associated with response to platinum-based chemotherapy (40% in responders vs. 7% in nonresponders) and overall survival (*p* = 0.049) [67]. This link was reported in another study by Van Allen et al., where ERCC2 somatic mutations were most commonly reported in responders to neoadjuvant treatment with cisplatin [68]. ERCC2 helicase domain mutations functionally impair NER pathway conferring cisplatin sensitivity to tumor cells [69]. In the metastatic setting, Teo et al. evaluated the presence of DDR somatic mutations using a 34-gene panel in patients with metastatic or advanced bladder cancer [70]. Somatic DDR alterations were associated with prolonged PFS (*p* = 0.007) and overall survival (*p* = 0.006) after platinum-based chemotherapy [70]. Regarding the BRCA1 DNA repair gene in urothelial cancer, Font et al. associated low BRCA1 expression with better response to neoadjuvant cisplatin-based chemotherapy (*p* = 0.01) and improved disease-free survival (*p* = 0.02) in patients with locally advanced bladder cancer [71]. A number of ongoing studies explore the role of DDR gene mutations in response to cisplatin in urothelial cancer. RETAIN is a Phase II study that enrolls patients with MIBC to undergo surgery or a bladder-preserving approach according to the mutational profile of the tumor (including ATM, RB1, FANCC and ERCC2 mutations) (NCT02710734) [72]. Another Phase 2 clinical trial is exploring the administration of dose-dense cisplatin and gemcitabine with bladder preservation in patients with MIBC-bearing deleterious DDR gene mutations (NCT03609216) [73]. In addition, DDR gene defections might confer susceptibility to drugs that interfere with DNA repair pathways, like PARP inhibitors. PARP inhibitors are currently under investigation in Phase 1/2 clinical trials in urothelial cancer. Patients with DDR gene alterations, like BRCA1 mutations, could benefit from the administration of PARP inhibitors. BISCAY multi-drug Phase 1b trial explores the administration of Durvalumab and Olaparib in patients with BRCA1/2, ATM or DDR gene alterations among other drug combinations (NCT02546661) [74]. ATLAS is a Phase 2 trial of Rucaparib PARP inhibitor in metastatic urothelial cancer (NCT03397394) [75]. The study was prematurely terminated due to limited efficacy. However, preliminary results of the trial showed no association between HRD status and response to rucaparib [76]. Discontinuation of treatment was reported in 95.9% of patients as progressive disease occurred in 73.1%. More results are anticipated. Another Phase 2 study is evaluating treatment with Olaparib in patients with advanced/metastatic urothelial cancer and DNA repair defects (NCT03375307) [77]. The study is under recruitment and is expected to finish in 2023. Finally, BAYOU is a Phase II study of Durvalumab plus Olaparib combination treatment versus durvalumab monotherapy as first-line therapy in patients with metastatic bladder cancer (NCT03459846) [78]. DDR gene mutations could be exploited by novel treatment strategies, revealing a new clinical significance in urothelial cancer.

## 10. DNA Repair Genes and Immune Checkpoint Inhibitors

Immunotherapy consists one of the major advances of bladder cancer treatment in the metastatic and recently in the adjuvant setting of high-risk disease. The role of ERCC1 expression or polymorphisms in response to checkpoint inhibitors is not well-defined. Most data emerge from studies evaluating DDR gene mutations as a whole in correlation with immunotherapy. Teo et al. explored the presence of DDR gene mutations in patients treated with atezolizumab or nivolumab for metastatic urothelial carcinoma [79]. DDR mutations were associated with higher response rate (67.9% vs. 18.8%; *p* < 0.001) to immunotherapy and improved PFS and OS rates [79]. Of note, mutational load was a predictor of response to treatment. A Phase 2 study of Gemcitabine/Cisplatin and Ipilimumab combination treatment was conducted in patients with metastatic urothelial cancer [80]. Galsky et al. reported that patients with deleterious DDR mutations exhibited higher response rates and improved but non-significant PFS and OS [80]. Of note, ATM gene mutations increased response to immunotherapy in bladder tumors by affecting the tumor microenvironment [81]. ATM mutations increased bladder cancer sensitivity to 29 drugs (*p* < 0.05), including cisplatin. Overall, DDR gene alterations increase susceptibility not only to platinum salts but also to immune checkpoint inhibitors, however, studies supporting this notion are still limited.

## 11. Discussion

Overall, ERCC1 expression may be exploited as both a prognostic and predictive biomarker in bladder cancer. Most studies agree on the favorable outcome of ERCC1-positive/high-expressing urothelial tumors that are not treated with platinum-based chemotherapy. Inversely, ERCC-negative/low-expressing tumors show a survival benefit compared with ERCC1-positive ones when platinum-based chemotherapy is administered in a neo-, adjuvant or metastatic setting. 

Indeed, a meta-analysis synthesizing results from six studies highlighted the significance of ERCC1 in platinum-treated urothelial cancer [82]. Low/negative expression of ERCC1 was associated with prolonged median PFS and OS in patients with bladder cancer receiving platinum-based chemotherapy [82]. This evidence would support the use of ERCC1 as a selection tool to distinguish urothelial cancer patients that will most benefit from neoadjuvant or adjuvant chemotherapy and optimize clinical practice. In NSCLC patients who did not receive platinum-based chemotherapy, ERCC1-positive/high expression has been also associated with a better outcome, in accordance with the aforementioned results [7].

There is a marked heterogeneity between study results assessing the predictive and prognostic role of ERCC1 in bladder cancer. This phenomenon is a result of multiple factors: different methods used for ERCC1 expression testing, differences in treatment regimens and dosing schedules applied, limited number of study subjects and, more importantly, a lack of standard threshold values for ERCC1 expression. Both IHC and RT-PCR have been used to evaluate ERCC1 creating an interstudy heterogeneity. The standard method to identify ERCC1 protein expression is through immunohistochemistry with monoclonal antibody 8F1, despite the development of novel antibodies [83]. Moreover, threshold values for the positivity and negativity of ERCC1 expression differ between studies. The application of a scoring system based on the relative expression of ERCC1 without accounting for the percentage of cells at each staining intensity level (e.g., H-score) remains controversial. Cisplatin resistance can emerge even from a small population of ERCC1 high-expressing tumor cells [83]. Another major obstacle to interstudy homogeneity is the expression of ERCC1 as four functionally distinct variants, namely isoforms 201, 202, 203 and 204 [84]. Friboulet et al. showed that isoform 202 is the sole one that is functionally active in terms of DNA repair [84]. ERCC1 isoform 202 and none of the other three variants is able to physically interact with XPF, RPA and XPA and rescue cell apoptosis due to replication stress, ICLs and DSBs. None of the available immunohistochemistry (IHC) anti-ERCC1 antibodies can efficiently distinguish ERCC1 isoforms, whereas no isoform-specific primers for PCR can be developed due to sequence homologies.

ERCC1 SNPs, mainly C118T (rs11615) and 17677A > C have been associated with bladder cancer occurrence and survival in several studies. These findings are in concordance with data observed in other solid tumors. We already stated the significance of ERCC1 SNPs as prognostic biomarkers in NSCLC, ovarian and colorectal cancer [8,40,42,45,46]. In breast cancer, ERCC1 C8092A (rs3212986) and C118T (rs11615) genotypes predicted a more favorable prognosis in locally advanced breast cancer treated with platinum-based chemotherapy [41]. Risk of glioma proved to be increased in the ERCC1 C8092A variant genotype [85]. ERCC1 C118T (rs11615) SNP was associated with increased risk of head and neck cancer in the overall population [9]. 

Collectively, a meta-analysis of forty-eight publications including 20,993 cases and 26,380 controls explored the contribution of ERCC1 SNPs to cancer susceptibility [86]. Both homozygote and heterozygote 17677A > C (rs3212961) variants were associated with significantly increased overall cancer risk. In addition, homozygote ERCC1 C118T (rs11615) variants were associated with lung and smoking-related cancer incidence [86]. However, results from different studies often provide contradicting results or statistically nonsignificant associations. An explanation for this phenomenon is the multifactorial nature of cancer. Individual SNPs may not be sufficient to produce a pathologic phenotype; rather, an additive effect of multiple SNPs in the same pathway is required for carcinogenesis. Only a pathway-based genotyping approach could combine the effects of individual SNPs and enhance predictive power.

There is a rising concern about nonresponding patients to platinum treatment who are exposed to substantial toxicity without receiving any clinical benefit. A pharmacogenomic or predictive biomarker is a molecule that correlates with drug efficacy or toxicity and informs on the appropriateness of a proposed therapy for a specific patient [87]. Once identified, nonresponders could avoid pointless systematic toxicity and receive an alternative more suitable therapy. Unfortunately, approximately one third of MIBC patients exhibit a pathologic response to platinum treatment [88,89]. Considering this low response rate, there is an unmet need for the identification of molecular biomarkers that could contribute to treatment optimization in urothelial cancer. Future studies are needed to prove if ERCC1 expression rates and gene polymorphisms could serve this purpose.

## Figures and Tables

**Table 1 ijms-21-08829-t001:** Expression with clinical outcome in the Neoadjuvant Setting.

Study	Sample Size	Treatment Regimen	Median DFS	Median OS	Pathologic Response	Method ERCC1 Detection	ERCC1 Positive/High-Expressing Tumors	Results	Statistical Significance
Ozcan et al. 2013 [22]	38	Gemcitabine-cisplatin, Gemcitabine-carboplatin, MVAC-MVEC	21.8	31	8/38 (21%) CR 9/38 (23.7%) CR/PR	IHC	9 (24%)	High ERCC1 expression—Short DFS High ERCC1 expression—Short OS	HR: 3.7; 95% CI: 1.2–11.2, *p* = 0.019 HR 6.1; 95% CI: 1.9–19.9; *p* = 0.002
Hemdan et al. 2014 [21]	244	Cisplatin-based chemotherapy	-	-	-	IHC	76%	No ERCC1 expression—Prolonged OS	HR: 1.77; 95% CI 1.22–2.56; *p* = 0.002
Eldehna et al. 2018 [23]	80	Gemcitabine-Cisplatin, Gemcitabine-Carboplatin	Lowest mean PFS 3.5 ± 0.71	Lowest mean OS 5 ± 0	37/80 (46.2%) CR 22/80 (27.5%) PR	IHC	35 (43.75%)	Negative ERCC1 expression—Increased treatment response (CR, PR) ERCC1 expression—Median PFS ERCC1 expression—Median OS	*p* = 0.013 *p*= 0.794 *p* = 0.499
Kawashima et al. 2012 [24]	58	Gemcitabine-Cisplatin, M-VAC	5-year 60.4%	5-year 66.3%	34% CR (20/58)	IHC	43% (25/58) 41% (24/58) ERCC1 & Snail positive	Positive ERCC1—Reduced DFS Positive ERCC1—Reduced DFS ERCC1 and Snail coexpression—Prolonged DFS ERCC1 and Snail coexpression—Prolonged OS	*p* = 0.055 *p* = 0.070 *p* = 0.029 *p* = 0.040
Necchi et al. 2017 [26]	46	Gemcitabine-Cisplatin-Sorafenib	2-years 73.3% (60.7–88.4)	2–years 82.8% (71.9–95.4)	53.4% (25/46) CR + PR	IHC	-	High ERCC1 expression—Decreased treatment response	*p* = 0.047
Choueiri et al. 2012 [27]		Dose-dense M-VAC	1-year 89% for responders 1-year 67% for non-responders		19/39 (49%) CR&PR 10/39 (26%) CR	IHC	12/31 (39%)	43% Negative ERCC1 expression 60% Positive ERCC1 expression—Pathologic response ERCC1 expression—DFS	Not significant
Yang et al. 2018 [28]	52	Gemcitabine-Cisplatin	-	-	75% (39/52) CR + PR	IHC	92% (48/52)	High ERCC1 expression- increased response to treatment	*p* = 0.011

DFS: disease -free survival; OS: overall survival; CR: complete response; PR: partial response; IHC: immunohistochemistry; M-VAC: methotrexate, adriamycin, vinblastine, cisplatin; MVEC: methotrexate, vinblastine, epirubicin, cisplatin; HR: hazard ratio; CI: confidence interval.

**Table 2 ijms-21-08829-t002:** Expression with clinical outcome in the Adjuvant Setting.

Study	Sample Size	Treatment Regimen	Median DFS	Median OS	Method ERCC1 Detection	ERCC1 Positive/High-Expressing Tumors	Results	Statistical Significance
Sun et al. 2012 [15]	57	Gemcitabine-Cisplatin	-	5-year 52.3%	IHC	34/57 (59.6%)	Positive ERCC1 expression—Shorter survival Negative ERCC1—Prolonged DFS Negative ERCC1—Prolonged OS	HR for death: 2.64; 95% CI 1.01–6.85; *p* = 0.047 *p* = 0.20 *p* = 0.034
Hoffman et al. 2010 [29]	108	Cisplatin-Methotrexate (CM) Methotrexate, Vinblastine, Epirubicin, Cisplatin (M-VEC)	-	-	RT-PCR	-	Low ERCC1 expression—Prolonged PFS Low ERCC1 expression—Prolonged OS	RR = 2.24; *p* = 0.01; 95%CI = 1.23–4.08 RR = 1.75, *p* = 0.10, 95% CI = 0.89–3.44
Klatte et al. 2015 [17]	432	Gemcitabine-Cisplatin M-VEC M-VAC	-	-	IHC	308 (71.3%)	ERCC1 expression—Treatment response	*p* = 0.88

DFS: disease -free survival; OS: overall survival; IHC: immunohistochemistry; RT-PCR: reverse transcription polymerase chain reaction; M-VAC: methotrexate, adriamycin, vinblastine, cisplatin; MVEC: methotrexate, vinblastine, epirubicin, cisplatin; HR: hazard ratio; CI: confidence interval.

**Table 3 ijms-21-08829-t003:** Expression with prognosis in the Locally Advanced/Metastatic Setting.

Study	Sample Size	Treatment Regimen	Median PFS	Median OS	Pathologic Response	Method ERCC1 Detection	ERCC1 Positive/High-Expressing Tumors	Results	Statistical Significance
Bellmunt et al. 2007 [30]	57	Gemcitabine-Cisplatin; Gemcitabine, Cisplatin, Paclitaxel	-	23.4 months (95% CI: 19.1–27.8 months) Low ERCC1: 25.4 months High ERCC1: 15.4 months	74.1% (CR/PR) 17 (31.5%) CR 23 (42.6%) PR	RT-PCR	20 (35.1%)	Low ERCC1 expression—Longer median OS Low ERCC1 expression—Prolonged PFS High ERCC1 expression—Worse prognosis	*p* = 0.03 *p* = 0.087 HR: 3.72; 95% CI 1.33–10.40; *p* = 0.012
Guix et al. 2009 [31]	51	Cisplatin-based	-	Median OS: 14.4 months (95% CI 6.7–16.1)		IHC	40% (21% weekly positive, 19% strongly positive)	Negative ERCC1 expression—Prolonged disease-specific survival (DSS)	(12.6 vs. 8.6 months) *p* = 0.032
Matsumara et al. 2010 [32]	40	Gemcitabine-Cisplatin, Gemcitabine, Cisplatin, Paclitaxel	10.7 months (95% CI: 7–14.4) Low ERCC1 expression: 13 months High ERCC1 expression: 10.7 months	15.5 months (95% CI: 12.4–18.6 months) Low ERCC1 expression: 13.6 months High ERCC1 expression: 17.1 months	62.5% (CR/PR)	IHC	19 (47.5%)	Median PFS—High/Low ERCC1 expression Median OS—High/Low ERCC1 expression High hENT1/Low ERCC1—Median OS vs. Low hENT1/High ERCC1—Median OS	(10.7 vs. 13 months) *p* = 0.33 (13.6 vs. 17.1 months) *p* = 0.178 19.9 months vs. 11.6 months
KH Kim et al. 2010 [33]	89	Gemcitabine-Cisplatin, MVAC	8.77 months (95% CI: 7.2–10.4)	25.2 months (95% CI: 17.5–32.8)	61 (68.5%) CR/PR 25 (28.1%) CR 36 (40.4%) PR	IHC	49 (55%)	Negative ERCC1 expression—Prolonged median PFS ERCC1 expression—Median OS	OR: 1.62; 95% CI: 1.03–2.54; *p* = 0.003 *p* = 0.73
Mullane et al. 2016 [34]	104	Platinum-based CT	-	-	-	IHC	-	Low ERCC1 expression—Prolonged OS Low ERCC1/RAD51 expression—Prolonged OS	HR = 2.7; *p* = 0.0007 *p* = 0.005
Song et al. 2016 [16]	42	Gemcitabine-Oxaliplatin	-	Positive ERCC1 expression: 16.1 months (95% CI: 11.72–11.72) Negative ERCC1 expression: 22.2 months (95% CI: 12.29–32.11)	-	IHC	20 (47.6%)	Negative ERCC1 expression—Longer OS Negative ERCC1/BRCA1 expression—Longer OS	*p* = 0.042 *p* = 0.018
Necchi et al. 2014 [35]	88	MVAC, CG, Carbo/Gem, CDDP/Gem/Paclitaxel, Carbo/Gem/Paclitaxel	10.3 months QR, 5.9–31.5 months	20.5 months (IQR, 11.7–44.5 month	-	IHC	30/66 (45.4%)	ERCC1 expression—Median PFS ERCC1 expression—Median OS	*p*= 0.750 *p*= 0.893
Kim et al. 2015 [19]	53	Gemcitabine-Cisplatin, Gemcitabine-Carboplatin	6.4 months	14 months	32 (60.4%) CR/PR	IHC	26 (49.1%)	Low/High ERCC1 expression—Response to treatment Low/High ERCC1 expression—Median PFS Low/High ERCC1 expression—Median OS	50% vs. 70.4% (*p* = 0.130) (8.8 vs. 5.8 months) *p* = 0.096 (17.2 vs. 12 months) *p* = 0.444
Urun et al. 2017 [36]	1475	Platinum-based CT	-	-	-	IHC/RT-PCR	-	Positive ERCC1 expression—Reduced PFS ERCC1 expression—Median OS ERCC1 expression—Median DFS	(HR: 1.54, 95% CI: 1.13–2.11, *p* = 0.006) HR1.63; 95% CI: 0.93–2.88; *p* = 0.09 HR: 1.092; 95% CI: 0.63–1.90; *p* = 0.75

**Table 4 ijms-21-08829-t004:** ERCC1 Single Nucleotide Polymorphisms (SNPs).

ERCC1 Polymorphisms
30-UTR C8092A (rs321986)
C19007T/118Asn (rs11615)
T1773C/154His
IVS3 + 74G > C
E × 3 − 97G > A/75Thr
IVS3 + 41G > T
IVS5 + 33C > A (rs3212961)
196 bp 3’ of STP G > T

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
