# Peer review of "Clinical Perspectives of ERCC1 in Bladder Cancer"

_ijms, 2020, doi:10.3390/ijms21228829_

Round 1

Reviewer 1 Report

could you please describe the methodology of selecting the articles for your review- which criteria were taken into account, how many articles were selected?

key words - breast cancer- is it necessary?

I would advise to shorten the manuscript considerably and concentrate mainly on clinical aspects of ERCC1,  you could also discuss this application of other potential markers of BCa

the title could be also more informative if possible i.e. clinical implications of ERCC1 in bladder cancer?

Reviewer 2 Report

Comprehensive nice review of the role of ERCC1 as a putative biomarker in bladder cancer. The review is relevant to broader biomarker discussions in the field and seems very detailed, including relevant literature. It is organized well, so it is easy to follow. Minor revisions suggested:

  1. Would mention a bit more about ERCC2, ATM, Rb1, FANCC, or other DDR gene mutations as potential predictors of response to cisplatin based chemotherapy, which are currently being in tested in 3 neoadjuvant trials
  2. Consider commenting on ERCC1 & other DDR genes and PARP inhibitors and briefly discuss/cite BISCAY trial https://oncologypro.esmo.org/meeting-resources/esmo-2019-congress/An-adaptive-biomarker-directed-platform-study-in-metastatic-urothelial-cancer-BISCAY-with-durvalumab-in-combination-with-targeted-therapies and ATLAS trial https://ascopubs.org/doi/abs/10.1200/JCO.2020.38.6_suppl.440 & 
  3. Line 379-380: ERCC1 has been studied extensively as the authors show but has not been useful yet in practice, so would rephrase that sentence
  4. Any comments about ERCC1 and response to checkpoint inhibitors?
  5. Avoid phrasing "cancer patients" but rather use "patients with cancer" as per patient advocacy groups
  6. Any comments about ctDNA as a tool complementary to tissue aiming to capture tumor heterogeneity and monitor alterations in ERCC1 gene?

Round 2

Reviewer 1 Report

thanks for the revised version of the manuscript